# The Contribution of Volatile Organic Compounds (VOCs) Emitted by Petals and Pollen to the Scent of Garden Roses

Matteo Caser and Valentina Scariot *

Department of Agricultural, Forest and Food Sciences, University of Torino, Largo Paolo Braccini 2, 10095 Grugliasco, Italy
* Correspondence: valentina.scariot@unito.it; Tel.: +39-011-6708932

**Abstract:** Flower scent is an important trait of ornamental roses and has been an important character in the selection processes. In the present study, the composition of the volatile organic compounds (VOCs) emitted by both petals and pollen of 21 garden roses (Chinensis, Climber, English rose, Floribunda, Hybrid Tea, Multiflora, Damascena, Musk rose, Polyantha, Rugosa and Shrub) was investigated through the GC-MS Static Headspace method. A total of 19 different VOCs were detected, and for each identified compound, an odorant description was included. In petals, the most common VOCs were 2-phenylethanol, methyl eugenol, and hexanal, present in 95%, 86% and 86% of garden roses, respectively. While, in pollen were methyl eugenol, methyl-1-butanol, and hexanal (present in 100%, 95%, and 90% of the genotypes, respectively), even if in lower content. The comparison between the petals and pollen profile shown that, even with less quantity, the main compounds characterizing the scent of the studied roses are present both in the petals and in the pollen (19 and 17 compounds, respectively), with different magnitude. Overall, the content of VOCs emitted by petals was more than five times higher than that produced by pollen. Different and characteristic VOCs profiles were emitted by petals and pollen of the studied garden roses.

**Keywords:** aroma active compounds; flower fragrance; odorant descriptor; relative peak area; *Rosa* L.; static headspace

## 1. Introduction

Plants synthesize and release a large variety of volatile organic compounds (VOCs), which can be considered as the plant's interface with its surrounding environment while remaining anchored to the ground [1]. VOCs have diverse chemical structures and are mainly classified into terpenoids, phenylpropanoids, and fatty acid derivates [2]. They can be constitutively emitted from flowers, leaves, fruits, roots, specialized storage structures, and other minor organs and tissues such as pollen [3,4]. In vegetative tissues, the volatile compounds render plant defense, whereas in reproductive tissues, they aid mainly in the attraction of pollinators [1]. VOCs play critical roles also in imparting cues to other plants to facilitate mating and adaptation to the changes in environment such as under biotic and abiotic stressed conditions [5–10].

Flower scent is an important trait of ornamental roses (*Rosa* L. spp.) that has provided pleasures for humans since antiquity, being used also for cosmetics, perfume industry, and medicine [11]. Currently, within this genus, about 250 species and more than 18,000 cultivars are documented [12]. However, among them, only few species are scented: *R. damascena* Mill., *R. gallica* L., *R. centifolia* L., *R. moschata* Herrm., *R. borboniana* N.H.F. Desp., *R. chinensis* Jacq., and *R. alba* L. [13–16]. Rose fragrance consists of more than 400 VOCs with diverse biosynthetic origins whose amounts vary among species and cultivars [14,15,17,18]. Rose's VOCs can be mainly grouped into terpenes, which are generally the most abundant compounds and consist of monoterpene alcohols and sesquiterpenes [19]; terpene derivatives, such as ketones, are present in extremely low quantities but which nevertheless

contribute significantly to the fragrance of the flower; lipid derivatives, that are synthesized by leaves and sepals in case of injury; and aromatic compounds such as 2-phenyl-ethanol, which can be very abundant and provide typical rose scent [1,15,20].

The amount and composition of VOCs is strongly affected by genotypes and technology used for processing [5,12]. Gas chromatography-mass spectrometry (GC-MS) is a powerful analytical technique that is widely used to profile volatile compounds in several plant species and food types because it can efficiently separate and identify compounds [12,21]. The ideal sample preparation method for the analysis of aroma compounds in rose is a headspace approach that involves the sampling of the gas phase above the sample in a closed container. Static headspace (SHS) sampling can be performed to extract VOCs from rose samples without the use of specific solid-phase materials [22]. After a certain range of time, the headspace gas is extracted from the vial and injected into a gas chromatograph, which separates the various components of the sample based on size and/or polarity. The resulting mass spectrum allows for the identification of the components using standard reference libraries. This method was already performed in several ambits, such as to sample the VOCs emitted from the leaves of *Corymbia citriodora* (Hook.) K.D. Hill & L.A.S. Johnson [23].

The literature on the chemistry of rose scent has extensively analyzed the VOCs emitted from petals and essential oils of a few species, commonly used in the perfume and cosmetic industry (*R. damascena*, *R. rugosa* Thunb., and *R. moschata*) [24–27]. Fragrances in garden roses are very diverse and scent has always been an important character in the selection process. Up to now, only old studies demonstrated that also the pollen of some botanical roses such as *R. rugosa* emitted volatile compounds [28,29] and very few studies evaluated the content and distribution of the VOCs emitted by both fresh petals and pollen on the same genotypes [28,30]. This study aimed to broaden the knowledge on garden rose fragrance by investigating the composition of the VOCs emitted by petals and pollen, through the SHS method, in 21 garden roses belonging from different classes such as Chinensis, Climber, English rose, Floribunda, Hybrid Tea, Multiflora, Damascena, Musk rose, Polyantha, Rugosa and Shrub. This research focused on describing a portion of the volatiloma of these genotypes and identify possible candidates for future breeding projects.

## 2. Materials and Methods

### 2.1. Plant Material

Fully opened flowers were picked from 21 garden rose genotypes (Figure 1 and Table 1) cultivated in the nursery "Vivaio Anna Peyron" located in Castagneto Po (Italy) (Lat. 45°9′36″72 N; Long.07°53′25″80 E; 200 m a.s.l.). Roses were classified according to Cairns [31] and the fragrance of flowers (+, mild; ++ moderate; +++ strong) were obtained based on the public website [32].

Petals (up to three grams) and pollen (up to one gram) from at least three different plants were separated and immediately ice stored in 20 mL headspace vials for further analysis.

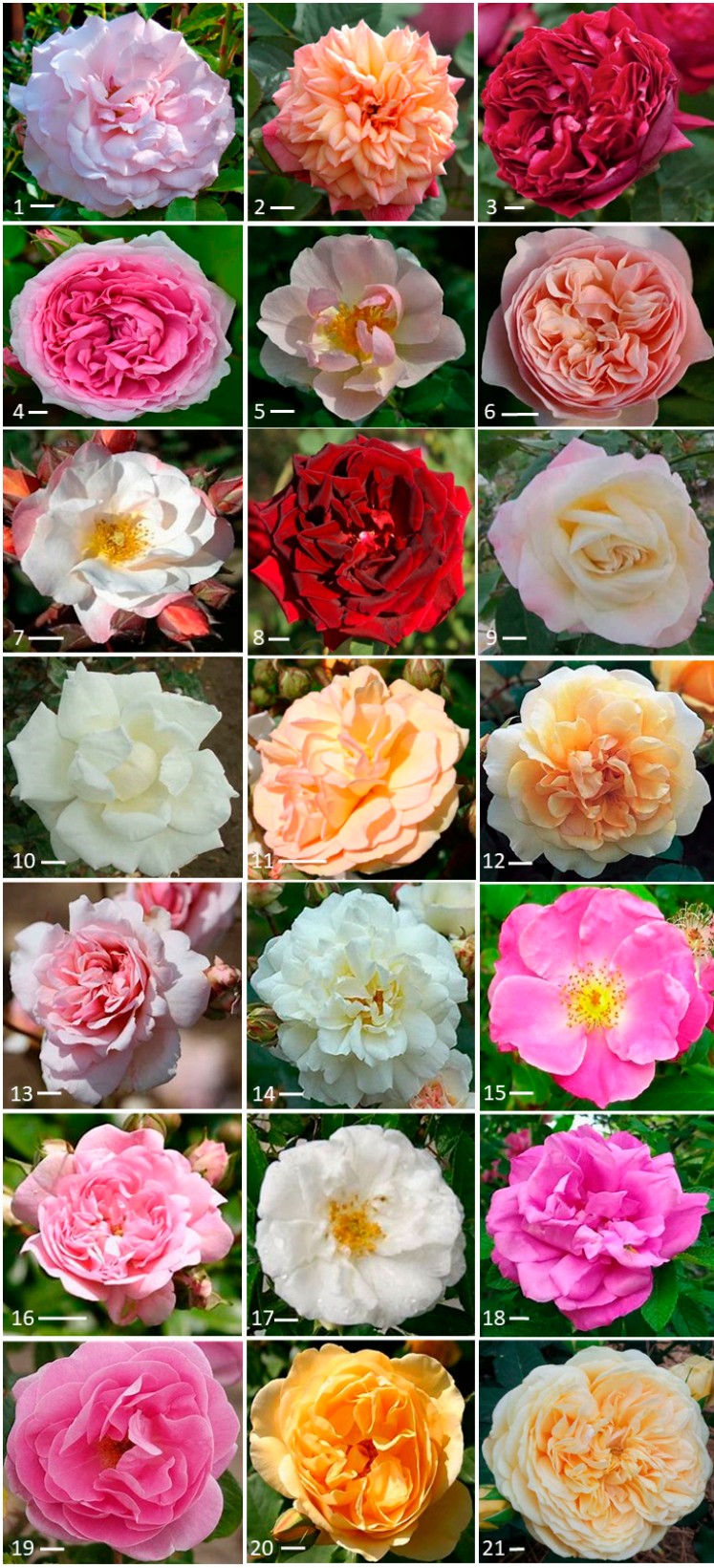

**Figure 1.** Flowers of the 21 studied garden rose genotypes (for the identification code see Table 1). The white scale bars within each figure indicate a 2 cm length.

**Table 1.** Identification number, cultivar, class [31], fragrance intensity (+, mild; ++ moderate; +++ strong) and type [32] of the tested garden roses (*Rosa* spp.).

| ID | Cultivar | Class | Fragrance Intensity | Fragrance Type |
|----|----------|-------|---------------------|----------------|
| 1 | 'Irene Watts' | Chinensis | ++ | Sweet |
| 2 | 'Aloha' | Climber | +++ | Apple |
| 3 | 'Eric Tabarly' | Climber | ++ | - |
| 4 | 'Comte de Chambord' | Damascena | +++ | Damask |
| 5 | 'Peach Blossom' | English rose | + | - |
| 6 | 'Sweet Juliet' | English rose | +++ | - |
| 7 | 'Clair Matin' | Floribunda | ++ | Sweetbriar |
| 8 | 'Crimson Glory' | Hybrid Tea | +++ | Clove, damask, rose |
| 9 | 'Marie Van Houtte' | Hybrid Tea | ++ | - |
| 10 | 'Mrs. Herbert Stevens' | Hybrid Tea | ++ | - |
| 11 | 'Ghislaine de Feligonde' | Multiflora | ++ | - |
| 12 | 'Buff Beauty' | Musk rose | +++ | Tea rose |
| 13 | 'Felicia' | Musk rose | +++ | Sweet |
| 14 | 'Prosperity' | Musk rose | ++ | - |
| 15 | 'Vanity' | Musk rose | ++ | Musk |
| 16 | 'The Fairy' | Polyantha | + | Apple |
| 17 | 'Yvonne Rabier' | Polyantha | ++ | - |
| 18 | 'Belle Poitevine' | Rugosa | ++ | Centifolia |
| 19 | 'Sarah Van Fleet' | Rugosa | +++ | Old rose |
| 20 | 'Graham Thomas' | Shrub | +++ | Tea rose |
| 21 | 'Sweet Caitlin' | Shrub | +++ | Anise, apricot, citrus, clove, myrrh, violets |

## 2.2. GC-MS Analysis

Gas chromatography (GC) analysis were performed by the laboratory of Ente Sviluppo Agricolo (Palermo, Italy), using an Agilent-technology chromatograph (Hewlett-Packard 5890 gas chromatography, Hewlett-Packard, Palo Alto, CA, USA) with HP-5 column (30 m × 0.32 mm i.d. × 0.25 μm) with automatic headspace injector (Thermo Fisher Scientific HS TriPlus 300Thermo Fisher Scientific, Waltham, MA, USA). The vial headspace (volume of 1.5 mL) containing volatile compounds from three grams of petals and one gram of pollen was directly introduced into a GC-MS system. The extraction incubation parameters were agitator oven temperature of 70 °C, agitation (250 times/min) and incubation of 7.00 min. The pressurization and injection needle time were set at 15 psi and 0.5 min, respectively.

The GC injector temperature and split ratio were set at 170 °C in split mode (10:1), respectively. The temperature program of the GC started at 40 °C, maintained for 1.00 min, then increased a rate of 5.0 °C/min to 170 °C, hold time 3.00 min, and then increased a rate of 5.0 °C/min to 200.0 °C, hold time 5.00 min. Ultrahigh purity helium (He) was used as a carrier gas with the flow rate at 1.00 mL/min and the ion source temperature and the interface temperature of the MS were at 200 °C. Detection was performed using electron ionization at 70 eV with a scan range of 30–300 *m/z* with an initial scan time of 1.00 min. A neural networks' in-house data acquisition, analysis, and visualization were assessed by Xcalibur 2.1 software (Xcalibur™, Finnigan Corp., Silicon Valley, CA, USA). Besides the possibility of quantifying single compounds, the purpose-made software allowed to match the peak pattern with sensory wise defined flavors. For each identified component, an odorant description was included based on previous publication released.

## 2.3. Calculation of the Percentage of the Relative Peak Area

The percentage of the relative peak area (%RPA) of a peak in each sample was calculated by dividing the peak area by the total peak area of all identified peaks in each chromatogram. The total ion chromatogram of each sample was used for peak area integration.

*2.4. Data Analysis*

Principal coordinate analysis (PCA)—Biplot was performed for both petal and pollen using PAST 4.03 software (Paleontological Statistic Software Package for Education And Data Analysis, Oyvind Hammer, Oslo, Norway). Eigenvalues were calculated using a covariance matrix among 38 traits as input (garden rose genotypes and emitted VOCs), and the two-dimensional PCA biplot was constructed. Radar charts were generated by using the rate between the %RPA of each odorant description and the sum of %RPA of airless headspace (Microsoft Excel for Windows, Office 365, Microsoft Corp., Redmond, WA, USA).

**3. Results**

*3.1. Identification of VOCs in Garden Rose Petals*

The main VOCs detected from garden rose petals are listed in Table 2. Their distribution in the 21 genotypes is shown in Table 3 and their odor description are represented in the related radar charts (Figure 2).

**Table 2.** List, retention time (R*t*), Kovats index (R*i*) and odorant description of the constituents identified both in petals and pollen of the 21 studied rose genotypes obtained by using the static headspace GC-MS method.

| Constituent | R*t* | R*i* | Odorant Description |
|---|---|---|---|
| Acetaldehyde | 1.53 | 358 [33] | Fruity, pungent [34] |
| Dimethyl sulfide | 1.63 | 505 [35] | Organic, wet earth [36] |
| Isovaleraldehyde | 2.45 | 632 [33] | Fruity [36] |
| 3-methyl-1-butanol | 3.35 | 736 [35] | Malty [34] |
| 1-pentanol | 3.90 | 772 [33] | Fruity [34] |
| Hexanal | 4.45 | 801 [35] | Sweet, green, apple [36] |
| Trans-2-hexenal | 5.49 | 854 [35] | Bitter, almonds, green, green apple-like, fatty, bitter almond like, cut grass [36] |
| Trans-3-hexen-1-ol | 5.61 | 873 [33] | Green [36] |
| Trans-2-hexen-1-ol | 5.95 | 879 [33] | Green [37] |
| Hexan-1-ol | 6.04 | 881 [33] | Fruity, aromatic, soft, cut grass [38] |
| Heptanal | 6.84 | 903 [35] | Fatty [39] |
| α-pinene | 7.73 | 939 [35] | Woody, coniferous [34] |
| Benzaldehyde | 8.73 | 960 [35] | Bitter almond [39] |
| *m*-cresol | 11.86 | 1084 [35] | Shoe polish, machine [40] |
| 2-phenylethanol | 14.89 | 1104 [33] | Floral, rose [41] |
| Nerol | 16.84 | 1233 [35] | Sweet, fruity, flower [42] |
| (E)-citral (Neral) | 18.77 | 1247 [35] | Lemon [34] |
| (Z)-citral (Geranial) | 19.21 | 1277 [35] | Sharp lemon, sweet [35] |
| Methyl eugenol | 20.11 | 1407 [35] | Spicy [34] |

**Table 3.** Chemical composition and the percentage of relative peak area (%) of volatiles emitted from petals of the 21 studied rose genotypes (see Table 1). All constituents are ordered on the basis of their retention time (see Table 2).

| Constituent | \| Genotype (ID) | | | | | | | | | | | | | | | | | | | | |
| --- | --- | --- | --- | --- | --- | --- | --- | --- | --- | --- | --- | --- | --- | --- | --- | --- | --- | --- | --- | --- | --- |
| | 1 | 2 | 3 | 4 | 5 | 6 | 7 | 8 | 9 | 10 | 11 | 12 | 13 | 14 | 15 | 16 | 17 | 18 | 19 | 20 | 21 |
| Air | 85.8 | 53.3 | 60.5 | 68.3 | 80.2 | 80.7 | 76.4 | 93.7 | 55.2 | 83.7 | 77.5 | 78.3 | 91.2 | 80.6 | 67.1 | 78.1 | 77.9 | 65.5 | 95.6 | 84.3 | 74.5 |
| Acetaldheyde | 9.9 | 6.1 | 9.8 | - | 9.2 | 8.1 | - | - | 6.0 | 9.0 | 8.7 | - | - | 15.9 | - | - | 12.9 | - | - | 9.1 | - |
| Dimethyl sulfide | - | - | - | 8.4 | - | - | 11.0 | - | - | - | - | 17.7 | - | - | 16.7 | 11.8 | - | 23.9 | - | - | 10.3 |
| Isovaleraldehyde | - | - | - | 0.4 | - | - | 0.5 | - | 0.2 | - | - | 0.1 | - | - | - | - | - | - | - | - | - |
| 3-methyl-1-butanol | 0.1 | - | - | 0.1 | 0.1 | 0.1 | - | 0.1 | 0.1 | 0.1 | - | 0.4 | 0.1 | 0.1 | 0.2 | 0.2 | 0.3 | 0.1 | - | 0.1 | 0.1 |
| 1-pentanol | 0.3 | - | 0.2 | - | 0.3 | 0.2 | 0.2 | 0.1 | 0.2 | - | 0.1 | 0.2 | 0.1 | - | - | 0.1 | 0.1 | 0.1 | 0.1 | 0.1 | - |
| Hexanal | - | 31.6 | 21.8 | 5.9 | 0.8 | 0.1 | 1.3 | 0.4 | 21.8 | 0.1 | 1.4 | - | 0.1 | 0.1 | 0.1 | 0.2 | 0.1 | - | 0.2 | 0.3 | 3.9 |
| Trans-2-hexenal | - | 0.1 | 0.1 | - | - | - | - | - | 0.1 | - | - | - | - | - | - | - | - | - | - | - | - |
| Trans-3-hexen-1-ol | - | 3.6 | 2.4 | 1.1 | - | - | - | - | 5.5 | - | - | - | - | - | - | - | - | - | - | - | 0.6 |
| Trans-2-hexen-1-ol | - | - | - | - | - | - | - | - | - | - | 8.5 | - | - | - | - | - | - | - | - | - | - |
| Hexan-1-ol | 0.4 | 0.3 | 0.6 | 0.9 | 3.4 | 0.8 | 0.1 | 1.3 | 0.7 | 0.7 | - | 0.3 | 0.5 | 0.8 | 1.6 | 1.6 | 0.7 | 0.5 | 1.1 | 1.5 | 0.1 |
| Heptanal | - | - | - | - | - | - | - | - | 0.1 | - | - | - | - | - | - | - | - | - | - | - | - |
| α-pinene | 1.1 | - | 0.1 | - | 0.1 | - | 0.1 | 0.3 | - | - | - | 0.1 | - | - | 2.3 | - | - | - | 0.2 | - | 0.1 |
| Benzaldehyde | - | - | 0.1 | 0.1 | - | - | 0.1 | 0.1 | 0.1 | - | - | - | - | - | - | - | - | - | - | - | 0.5 |
| *m*-cresol | - | - | - | 0.1 | - | - | 0.1 | - | 0.1 | - | - | - | - | - | 0.1 | - | 0.1 | 0.1 | - | 0.1 | 0.1 |
| 2-phenylethanol | 0.5 | 0.3 | 2.9 | 11.5 | 1.8 | 7.9 | 7.8 | 2.4 | 8.7 | 4.0 | 0.6 | 1.2 | 6.4 | 1.5 | 6.2 | 5.8 | 5.6 | 5.2 | 1.4 | - | 4.2 |
| Nerol | - | - | - | - | - | 0.1 | 0.1 | - | - | 0.2 | - | - | 0.1 | - | 0.7 | 0.2 | - | 1.9 | - | 0.4 | 0.9 |
| (E)-citral (neral) | - | - | - | - | - | - | - | - | - | 0.1 | - | - | - | - | 0.1 | - | - | 0.1 | - | 0.1 | 0.1 |
| (Z)-citral (geranial) | - | - | - | 0.4 | - | - | 0.1 | - | - | - | - | 0.1 | 0.1 | - | - | - | - | 0.1 | - | 0.1 | 0.1 |
| Methyl eugenol | 0.1 | - | 0.1 | 0.1 | 0.1 | 0.1 | 0.1 | 0.1 | - | 0.1 | 0.1 | 0.1 | 0.1 | 0.1 | - | 0.1 | 0.1 | 0.1 | 0.1 | 0.1 | 0.1 |
| Total | 98.3 | 95.4 | 98.6 | 97.3 | 96.0 | 98.1 | 97.8 | 98.3 | 98.5 | 98.0 | 96.8 | 98.4 | 98.5 | 99.0 | 95.1 | 97.8 | 97.9 | 97.5 | 98.7 | 96.2 | 95.6 |

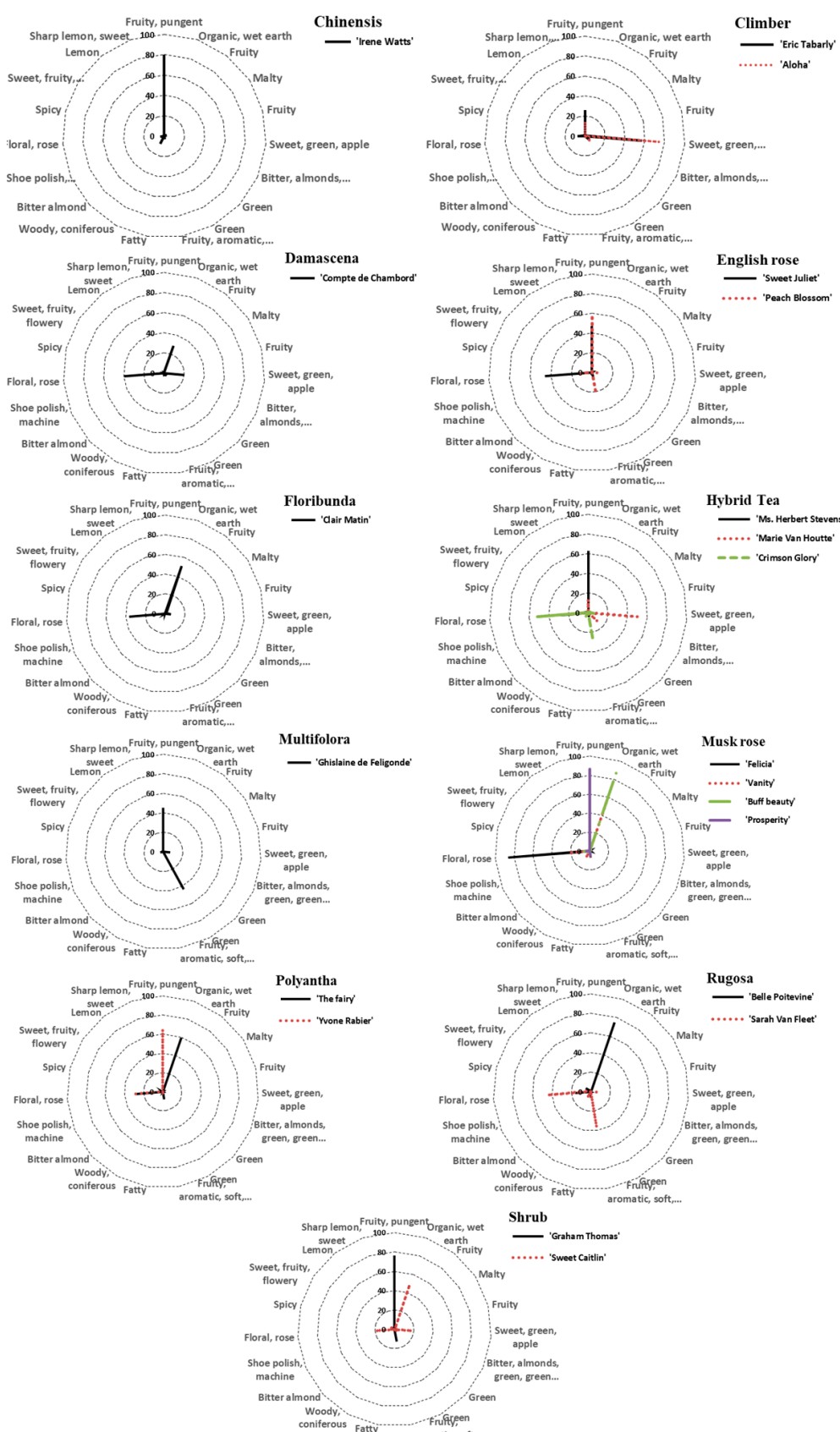

**Figure 2.** Radar charts showing the percentage (0–100%) of each odour description (see Table 2) within the airless headspace of the petals of the 21 studied garden rose genotypes. Cultivars were grouped in horticultural classes, as indicated in Table 1.

In general, a part air, a total of 19 different VOCs were detected. The most common were 2-phenylethanol, methyl eugenol, and hexanal, present in 95%, 86% and 86% of roses, respectively (Table 3). Among them, methyl eugenol was always present in very low concentration (average of RPA equal to 0.1%), while the others at superior content, ranging between 0.1% and 31.6% RPA and 0.3% and 11.5% RPA for hexanal and 2-phenylethanol, respectively. Among the remaining 17 compounds, three of these, hexan-1-ol, acetaldehyde, and dimethyl sulphide, are present in at least 33–52% of the studied varieties.

The cultivar with more identified compounds was 'Sweet Caitlin' with 13 molecules, while the cultivar with the highest content of molecules was 'Marie Van Houtte' with a total of RPA percentage equal to 43.3% (with 55.2% of air) (Table 3).

The headspace of the studied petals generally presented a distinctive molecule profile (Figure 2). The cultivars 'Irene Watts', 'Peach Blossom', 'Sweet Juliet', 'Mrs. Herbert Stevens', 'Ghislaine de Feligonde', 'Prosperity', 'Yvonne Rabier', and 'Graham Thomas' have acetaldheyde and a pronounced fruity and pungent like odor. The cultivars 'Clair Matin', 'Buff Beauty', 'Vanity', 'The Fairy', 'Belle Poitevine', and 'Sweet Caitlin' were characterized by dimethyl sulphide and an organic and wet like odor. Instead, hexanal is the most present compound in the VOCs emitted by the petals of the cultivars 'Aloha', 'Eric Tabarly', and 'Marie Van Houtte' with a sweet, green and apple like odor. Lastly, 2-phenylethanol and a related floral and rose like odor described the cultivar 'Comte de Chambord', 'Crimson Glory', 'Felicia', and 'Sarah Van Fleet'.

### 3.2. Identification of VOCs in Rose Pollen

The VOCs detected in the studied rose pollen are reported in Tables 2 and 4. In total, a part air, the applied SHS method allowed to identify 17 compounds. The cultivar 'Belle Poitevine' presented the highest number of detected compounds (12) with a total amount of 16.1 RPA%. The most common VOCs are methyl eugenol, 3-methyl-1-butanol, and hexanal (present in 100%, 95%, and 90% of the cultivars, respectively), even if in low concentration (0.1, 0.6 and 0.5 RPA%, respectively). Different compounds resulted distinctive of some varieties as shown by the radar charts in Figure 3. Acetaldheyde, with its fruity, pungent like odor, is the main compounds of the pollen of the cultivars 'Peach Blossom' and 'Crimson Glory'. Dimethyl sulphide, with its organic and wet earth like odor, characterized the cultivar 'Belle Poitevine'. 2-phenylethanol and the related floral and rose like odor characterized the cultivar 'Comte de Chambord', 'Prosperity', and 'Yvonne Rabier'. The odor of seven roses ('Aloha', 'Clair Matin', 'Marie Van Houtte', 'Ghislaine de Feligonde', 'Vanity', 'Buff Beauty', 'The Fairy' and 'Sweet Caitlin') was mainly described as malty due to 3-methyl-1-butanol. Instead, hexanal is the most present compound in the VOCs emitted by the pollen of the cultivars 'Irene Watts', 'Sweet Juliet', 'Mrs. Herbert Stevens' and 'Graham Thomas' with a sweet, green and apple like odor. Lastly, pollen of 'Eric Tabarly', 'Felicia' and 'Sarah Van Fleet' was mainly typified by methyl eugenol (spicy), hexan-1-ol (fruity, aromatic, soft, cut grass) and nerol (sweet, fruity, flower).

**Table 4.** Chemical composition and the percentage of relative peak area (%) of volatiles emitted from pollen of the 21 studied rose genotypes (see Table 1). All constituents are ordered on the basis of their retention time (see Table 2).

| Constituent | Genotype (ID) | | | | | | | | | | | | | | | | | | | | |
|---|---|---|---|---|---|---|---|---|---|---|---|---|---|---|---|---|---|---|---|---|---|
| | 1 | 2 | 3 | 4 | 5 | 6 | 7 | 8 | 9 | 10 | 11 | 12 | 13 | 14 | 15 | 16 | 17 | 18 | 19 | 20 | 21 |
| Air | 97.6 | 93.6 | 99.1 | 92.3 | 83.7 | 83.4 | 93.0 | 84.0 | 92.6 | 95.8 | 97.6 | 94.0 | 94.6 | 97.4 | 92.4 | 97.5 | 95.8 | 80.6 | 97.9 | 93.8 | 94.2 |
| Acetaldheyde | - | - | - | - | 11.4 | - | - | 8.4 | - | - | - | - | - | - | - | - | - | - | - | - | - |
| Dimethyl sulfide | - | - | - | - | - | - | - | - | - | - | - | - | - | - | - | - | - | 12.2 | - | - | - |
| Isovaleraldehyde | 0.3 | - | - | - | - | - | 0.5 | - | 0.1 | - | - | 0.3 | 0.2 | - | 0.7 | - | - | 0.4 | - | - | - |
| Methyl-1-butanol | 0.1 | 0.9 | - | 1.0 | 0.5 | 0.1 | 1.2 | 0.3 | 0.7 | 0.2 | 0.3 | 1.0 | 0.5 | 0.2 | 1.6 | 0.2 | 0.2 | 0.3 | 0.1 | 0.3 | 1.5 |
| 1-pentanol | - | 0.1 | - | - | 0.2 | - | 0.1 | - | - | - | - | - | - | - | 0.1 | - | - | - | - | - | - |
| Hexanal | 0.5 | 0.2 | - | 0.3 | 0.3 | 4.3 | 1.1 | 0.1 | - | 0.2 | 0.1 | 0.3 | 0.2 | 0.2 | 0.1 | 0.1 | 0.2 | 0.3 | 0.1 | 0.7 | 0.4 |
| Trans-2-hexenal | - | 0.1 | - | 0.1 | - | - | 0.1 | - | - | - | - | - | - | - | - | - | - | - | - | - | - |
| Hexan-1-ol | 0.1 | 0.3 | - | 0.2 | 0.6 | 1.2 | 0.1 | - | - | 0.1 | - | - | 0.5 | 0.1 | 0.3 | 0.1 | 0.1 | 0.2 | - | 0.1 | 0.1 |
| Heptanal | - | - | - | - | - | - | - | - | 0.1 | - | - | - | - | - | - | - | - | 0.1 | - | - | - |
| α-pinene | - | - | - | 0.2 | - | 0.1 | - | - | - | - | - | - | - | - | - | 0.2 | - | - | - | - | - |
| Benzaldehyde | - | - | - | - | - | - | 0.1 | - | - | - | 0.1 | 0.1 | - | - | - | - | - | 0.1 | - | - | - |
| *m*-cresol | - | - | - | - | - | - | - | - | - | - | - | - | - | - | - | - | - | - | - | - | 0.1 |
| 2-phenylethanol | - | 0.2 | - | 1.7 | 1.3 | 0.2 | - | - | - | 1.1 | - | 0.7 | 0.7 | 0.2 | 0.2 | - | 1.1 | 1.9 | - | - | - |
| Nerol | 0.1 | - | - | 0.7 | - | 2.2 | - | 0.3 | - | - | - | - | 0.2 | - | - | - | - | 0.4 | 0.1 | - | - |
| (E)-citral (neral) | - | - | - | 0.2 | - | 1.2 | - | 0.1 | - | - | - | - | 0.1 | - | - | - | - | 0.1 | 0.1 | - | - |
| (Z)-citral (geranial) | - | - | - | 0.2 | - | 1.7 | - | 0.1 | - | - | - | - | 0.1 | - | - | - | - | 0.1 | - | - | - |
| Methyl eugenol | 0.1 | 0.1 | 0.1 | 0.1 | 0.1 | 0.1 | 0.1 | 0.1 | 0.1 | 0.1 | 0.1 | 0.1 | 0.1 | 0.1 | 0.1 | 0.1 | 0.1 | 0.1 | 0.1 | 0.1 | 0.1 |
| Total | 98.8 | 95.5 | 99.2 | 96.8 | 98.2 | 94.2 | 96.2 | 93.2 | 93.6 | 97.7 | 98.2 | 96.4 | 97.3 | 98.3 | 95.3 | 98.2 | 97.7 | 96.7 | 98.5 | 95.0 | 96.3 |

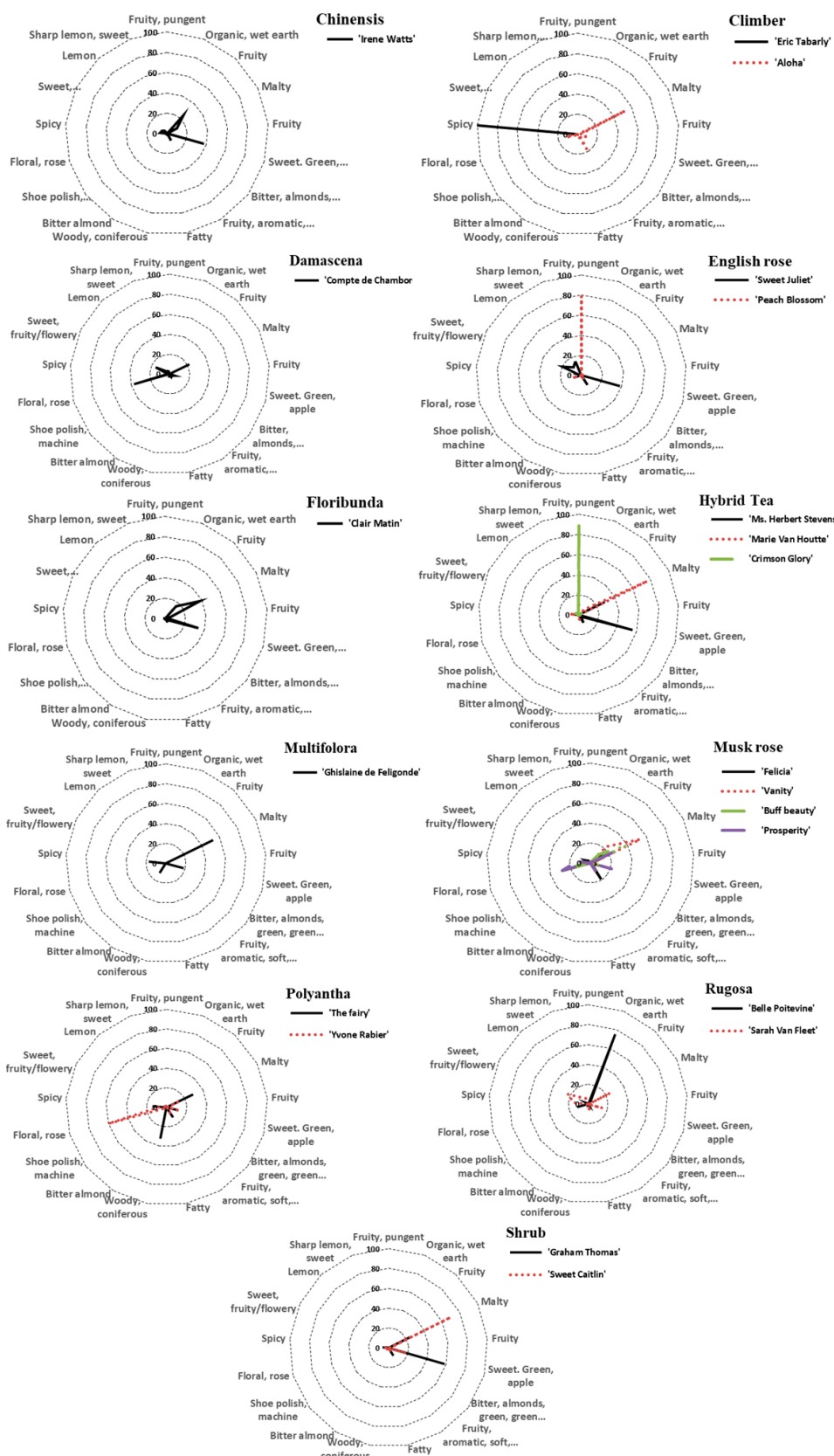

**Figure 3.** Radar charts showing the percentage (0–100%) of each odor description (see Table 2) within the airless headspace of the pollen of the 21 studied garden rose genotypes. Cultivars were grouped in horticultural classes, as indicated in Table 1.

### 3.3. Principal Component Analysis (PCA)

Based on Mantel tests, no correlation between the petal and pollen metabolite profiles were found (Mantel r statistic = −0.19, *p* = 0.75). In order to visualize congruence between garden rose genotypes and emitted volatile metabolites in petal and pollen, the whole datasets were subjected to a Principal Component Analysis (PCA; Figures 4 and 5). Three main groups related to petal VOCs were clearly divided by the first two axes (36.4% and 26.1% of the variance accounted for Axis 1 and Axis 2, respectively) (Figure 4).

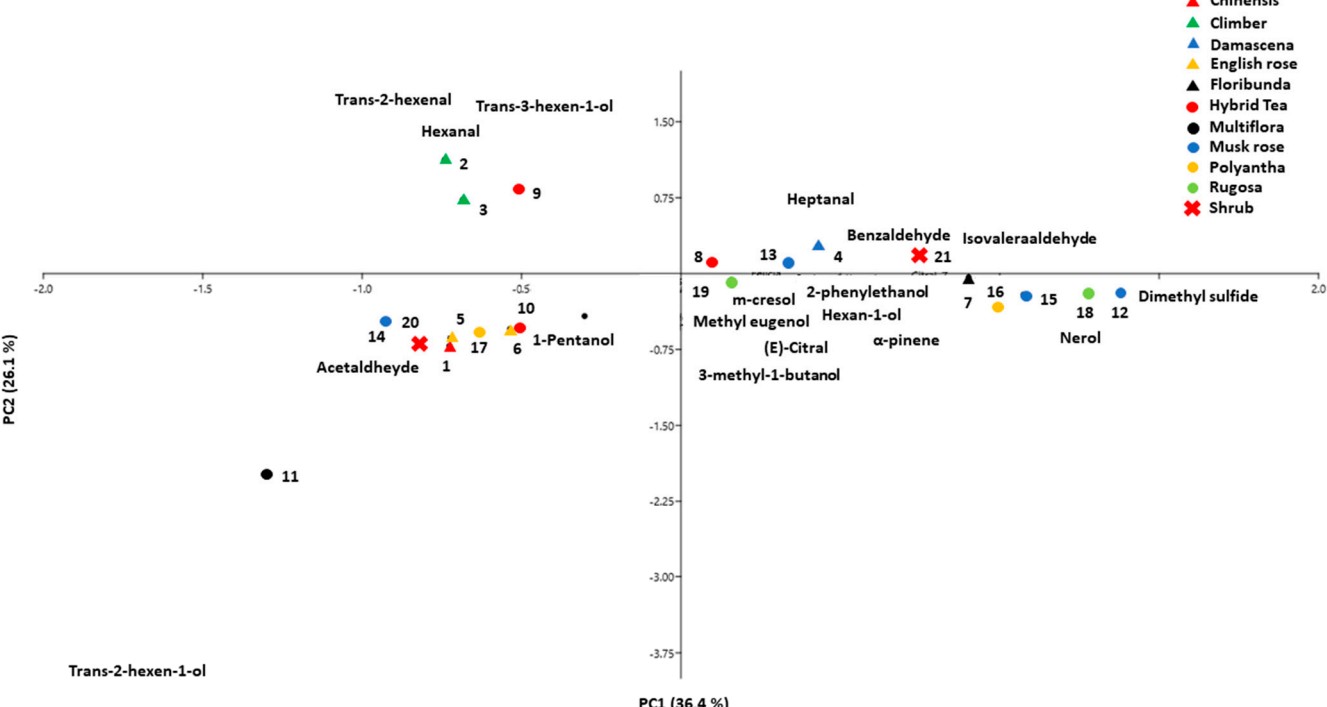

**Figure 4.** Principal component analysis (PCA)-biplot of the studied garden rose genotypes and volatile compounds emitted by petals. Numbers indicated the garden rose genotypes as listed in Table 1.

The two climber cultivars 'Aloha' and 'Eric Tabarly' and the Hybrid Tea 'Marie Van Utte' were grouped for negative values of Axis 1 and positive values of Axis 2. These cultivars resulted positively correlated with the emission of hexanal, trans-2-hexenal, and trans-3-hexen-1-ol. At negative values of both axes were mainly grouped the cultivars belonging to the English rose, Chinensis, Multiflora ('Peach Blossom', 'Sweet Juliet', 'Irene Watts', 'Ghislaine de Feligonde', respectively), and from other classes such as 'Graham Thomas', 'Yvonne Rabier', 'Mrs. Herbert Stevens', and 'Prosperity'. These cultivars resulted correlated to the emission of trans-2-hexen-1-ol, acetaldehyde, and 1-pentanol. All the other cultivars and volatiles clustered for positive values of Axis 1. Between them, 'Compte de Chambord' and 'Sweet Caitlin' resulted positively correlated to heptanal, benzaldehyde and isovaleraldehyde. The PCA based on rose pollen data set is shown in Figure 5.

Three main groups were clearly potted by Axis 1 and Axis 2, which accounted for 32.3% and 25.7% of variance, respectively. The cultivars 'Peach Blossom' and 'Crimson Glory' grouped together for negative values of Axis 1 and positive of Axis 2. The headspace emitted by their pollen was positively related to acetaldehyde. While, for positive values of both axes, the cultivar 'Belle Poitevine' was plotted closed to dimethyl sulfide, heptanal, and 2-phenylethanol. All the remained cultivars grouped together for positive values of Axis 1 and negative of Axis 2 and resulted related to the remained volatile compounds.

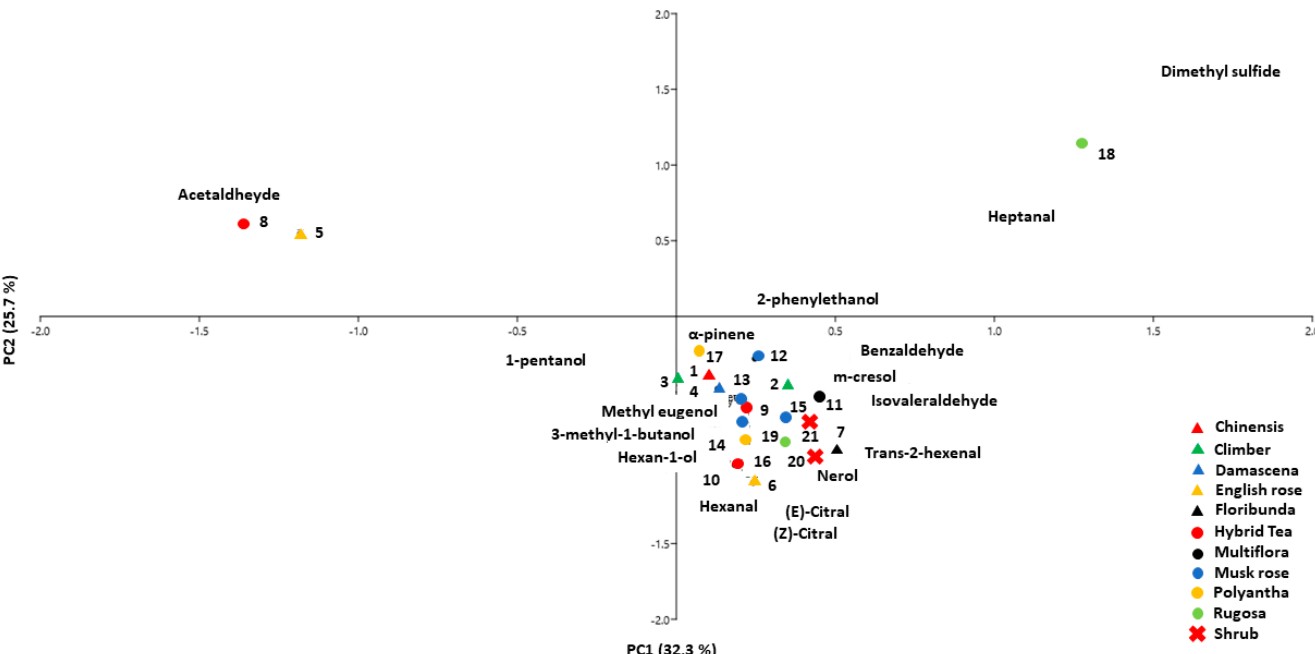

**Figure 5.** Principal component analysis (PCA)-biplot of the studied garden rose genotypes and volatile compounds emitted by pollen. Numbers indicated the garden rose genotypes as listed in Table 1.

## 4. Discussion

Roses are called the "queen of scents" because of the attractive and sweet aroma that they emit [1]. Floral scent is a composite character determined by a complex mixture of low-molecular-weight volatile molecules [43]. Rose breeders have recently tried to introduce new fragrances, for instance, reminiscent of fruit or spice odors. But despite their efforts, some roses in the market are not very fragrant, especially the ones selected for the cut flower market. The cause of this lack of scent is not completely known [44].

Regarding petals, the applied method was able to identify 19 molecules with methyl eugenol, hexanal, and 2-phenylethanol as the most present. Among them, methyl eugenol is reported as a naturally occurring carcinogenic compound found in a number of essential oils, including rose oil distilled from *Rosa damascena* Mill flowers [45]. As reported by Hirata et al. [1], 2-phenylethanol is generally emitted by petals of European roses that are used for perfume industries. In according with our results, Feng et al. [25], by using a GC-MS method, identified a total of 33 volatile compounds from 23 *R. rugosa* cultivars, with 2-phenylethanol as one of the major components of the petals. This compound was present also in the headspace of *R. hybrida* 'Honesty' [24]. More recently, Jandoust and Karami [27], applying a CombiPAL Headspace Techniques to evaluate the floral scent from fresh flowers of *R. moschata*, identified 21 VOCs with 2-phenylethanol as the most important component. This compound is known to be an insect attractant for seed dispersers [29,46]. Regarding the other detected compounds, Ibrahim et al. [47] by using a headspace solid phase micro-extraction identified a total of six and 14 volatile compounds in petals of different cultivars of Floribunda and Hybrid Tea roses, respectively, with α-pinene as a main compound. Accordingly, we observed the presence of α-pinene in the studied Floribunda rose 'Clair Matin'. At the full-blooming stage, Ren et al. [48] identified 65 volatile compounds from the petals of the cultivar 'Rose de Rescht', a damask rose. These authors detected as main constituent nerol as well as in the cultivar 'Comte de Chambord', in which we detected a high content of nerol. These components contribute mainly to the perfumery value of rose oil and are generally present in the distillate rose water [49–51].

The characterization of chemical constituents present in the odor of pollen has been largely ignored because of difficulties in sampling and analysis. Even if petal epidermal

are the main sites for scent production and emission [48–53], few and old studies demonstrated that also the pollen of some botanical roses such as *R. rugosa* emitted volatile compounds [28,29]. This agrees with our results on *R. rugosa* 'Belle Poitevine', that showed a pollen volatile profile with dimethyl sulfide as main compound. Dobson et al. [28] reported that in this species, the androecium and particularly the pollen is a major source of scent. In *R. rugosa*, pollen-specific volatiles are used by foraging bumblebees to assess the availability of pollen in flowers [29]. Emission of this volatile is predicted to be the key olfactory cue of *Satyrium pumilum* Thunb. flowers for attracting the flesh-eating fly pollinators [54], as assessed with bioassays in other plants such as *Helicodiceros muscivorus* (L. f.) Engl. [55–57]. Some VOCs in pollen have concurrent multiple functions. A variety of volatiles identified in the scent and considered to be attractants to pollinators might also have microbial and fungal defensive functions [58]. In the studied pollens, methyl eugenol was detected in all the genotypes and this molecule is reported to have antimicrobial activities that could protect pollen [59].

Samples of odors from pollen analyzed with headspace techniques have been found to be chemically different from scent from whole flowers [60,61], and the diversity of compounds identified is often lower in pollen [61,62]. The comparison between the petals and pollen profile shown that, even with less quantity, the main compounds characterizing the scent of the studied roses are present both in the petals and in the pollen (19 and 17 compounds, respectively), with different magnitude. Overall, the content of volatiles emitted by the petals was considerably higher than that produced by pollen (more than five times), suggesting that contributed most to the whole-flower fragrance. Odors from pollen are probably detected at short distances by insects in those cases in which pollen emissions of VOCs are quantitatively less abundant than those from the entire flower [59]. However, many species may present stronger pollen odors than others [63]. Among the identified volatiles, 2-phenylethanol was detected in both petals and pollen in most of the studied cultivars. To the best of our knowledge, this is the first time in which this molecule was identified not only in petals but also in pollen. Specifically, this molecule is present in high content in both the studied *R. rugosa* cultivars 'Belle Poitevine' and 'Sarah Van Fleet'.

No correlation between petal and pollen data sets was observed, thus we can presume that the biosynthesis of the VOCs follows different biosynthetic pathways. These findings agree with those obtained by Dobson et al. [28,30] in whole flowers and pollen of *R. rugosa* and *R. canina* L. Dobson et al. [28] highlighted that the distinct fragrances that characterize the different flower parts of *R. rugosa* differently contribute for distinctive roles: petals to attract pollinators, pollen to signal food availability, androecium to guide pollinators and sepal to protect from herbivores. An example is found in foraging bumblebees, in which landing is most effectively elicited when combining olfactory signals from pollen with visual stimuli from anthers [64]. Dobson et al. [29], in a series of behavioral field studies of bumblebees foraging for pollen on *R. rugosa*, provided the strongest evidence that bees use scents from pollen to distinguish between flowers that have different amounts of pollen. Similar results were observed also in the emission of volatiles from different organs of *Inula viscosa* (L.) Aiton and *Capparis spinosa* L. [65,66].

Apart from the ecological functions, garden roses are selected primarily for fragrance, whereas marketed roses bred for cut flower production often lack perfume, notwithstanding the efforts of breeders [67]. As recently reported by Baudino et al. [44] and Giovannini et al. [68], even fragrance-free roses emit small quantities of fragrant molecules, as they have not completely lost their ability to produce them, but according to a recent hypothesis, it would be a malfunction of the biosynthetic pathway of VOCs. Unfortunately, the biosynthetic pathways of many rose scent compounds are not completely known. On this topic, Yan et al. [69] suggested that a eugenol synthase (RcEGS1) cloned from the petals of *R. chinensis* 'Old Blush' is involved in the biosynthesis of methyl eugenol in rose, studying its over expression and down regulation by gene silencing. Taking together all of this information, our work can be of precious interest in the development of new

technologies in the field of breeding, suggesting the presence of molecules present even in low quantities both in the petals and in the pollen of limited studied garden roses.

## 5. Conclusions

The conducted analyses have shown that different and characteristic VOCs profiles are emitted by petals and pollen of the studied garden roses. The obtained dataset could become a valuable resource for the floricultural industry as well as for the fragrance, cosmetic, and food industries.

The static headspace method can be used as both a quality assessment system and a mean to distinguish rose varieties by using also scarce compounds emitted at low rates. Therefore, volatile compounds determined using this technique might serve as flavor tools for breeders in improving sensory quality. Some studies have confirmed that the functions of different floral organs are reflected in different profiles of emission. A desirable long-term goal of this research is to determine composition and function of floral scent bouquets on a finer spatial scale within the flower and their biosynthetic pathways.

**Author Contributions:** Conceptualization, V.S.; experimental set up, M.C. and V.S.; data analysis, M.C.; writing-original draft preparation, M.C.; writing-review and editing, M.C. and V.S.; supervision, V.S. All authors have read and agreed to the published version of the manuscript.

**Funding:** This research received no external funding.

**Acknowledgments:** The authors thank Andrea Berruti and Saskia Pellion di Persano for their contribution in the set-up of the trials, Renato Bosco, and Ente Sviluppo Agricolo of Palermo for the contribution in the data detection and analysis.

**Conflicts of Interest:** The authors declare no conflict of interest.

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
