# Peer review of "The Contribution of Volatile Organic Compounds (VOCs) Emitted by Petals and Pollen to the Scent of Garden Roses"

_horticulturae, doi:10.3390/horticulturae8111049_

Round 1
Reviewer 1 Report
In this paper, Caser et al., measured the volatile organic compounds (VOCs) in petals and pollen of 21 garden roses, and analyzed the correlation between rose genotypes and VOCs. The data in this study is quite comprehensive. Below are my comments.
1. Please indicate the number of samples or biological replicates.
2. Line 169: 2-phenylethanol should account for 95% of roses. 2-phenylethanol 20/21=95, methyl eugenol 18/21=86
3. The scale bars should be indicated in Figure 1.
Author Response
October 10th, 2022
To:
Editorial Office
Horticulturae
Dear Editorial Office,
We submit a revised version of the article “The contribution of volatile organic compounds (VOCs) emit-ted by petals and pollen to the scent of garden roses” by Matteo Caser and Valentina Scariot for publication in the journal “Horticulturae”.
We thank for the comments and suggestions that were very helpful to further improve clarity of the manuscript.
For the preparation of the revised manuscript, we followed all the comments and suggestions of the editor and the reviewers as stated below. We highlighted the main changes to the text by red.
Reviewer 1
Point 1: In this paper, Caser et al., measured the volatile organic compounds (VOCs) in petals and pollen of 21 garden roses, and analyzed the correlation between rose genotypes and VOCs. The data in this study is quite comprehensive. Below are my comments.
Point 1: we thank the Reviewer.
Point 2: Please indicate the number of samples or biological replicates.
Point 2: we added more information in M&M section.
Point 3: Line 169: 2-phenylethanol should account for 95% of roses. 2-phenylethanol 20/21=95, methyl eugenol 18/21=86
Point 3: we modified the text, accordingly.
Point 4: The scale bars should be indicated in Figure 1.
Point 3: we modified Figure 1, accordingly.
Best regards,
Matteo Caser and
Valentina Scariot
Department of Agricultural, Forest and Food Sciences
University of Turin
Largo Paolo Braccini, 2
10095, Grugliasco (TO)
Italy
Phone number: +039-011/6708935
Fax number: +039-011/6708798
e-mail: matteo.caser@unito.it

Reviewer 2 Report
The work is interesting, but it contains many scientific and editorial errors and cannot be published in its current form. The text should be corrected by a chemist. The main weaknesses are listed below:
- line 11, 73, the authors in the text write the Multiflora rose twice and omit Damascena.
- the authors use incorrect names of chemical compounds in the text, tables and figures. It is “exanal” and it should be “hexanal”, it is “trans-2-exenal” and it should be “trans -2-hexenal”, it is “trans-3-exen-1-ol” and it should be “trans-3-hexen-1-ol”, it is “trans-2- exen-1-ol” and it should be “trans-2-hexen-1-ol”, there is “alfa-pinene” and it should be “α-pinene”, there is “Citral_E” and it should be “(E)-citral”, there is “Citral_Z” and it should be “(Z)-citral”, is “3, methyl-1-butanol” should be “3-methyl-1-butanol” is “2-phenyethanol” and it should be “2-phenylethanol”
- table 2. The authors give incorrect references to chemical compounds. For example, in publication 32 we will not find information about acetaldehyde, 3-methyl-1-butanol, 1-pentanol, (Z)-citral, (E)-citral, and in publication 34 we will not find information about trans-2-hexen-1ol. Identification of compounds on the basis of retention time only is insufficient, the values of the Kovats retention indices must be included in table 2.
- chapter 2.2 does not contain significant information about the performed GC-MS analysis: chromatograph model, headspace automatic injector model, amount of sample injected, mass spectrometer parameters, MS library and so on. The given column temperature program is incorrect, it shows that the end is after 15 minutes and methyl eugenol, nerol, (E)-citral, (Z)-citral have longer retention times.
Author Response
October 10th, 2022
To:
Editorial Office
Horticulturae
Dear Editorial Office,
We submit a revised version of the article “The contribution of volatile organic compounds (VOCs) emit-ted by petals and pollen to the scent of garden roses” by Matteo Caser and Valentina Scariot for publication in the journal “Horticulturae”.
We thank for the comments and suggestions that were very helpful to further improve clarity of the manuscript.
For the preparation of the revised manuscript, we followed all the comments and suggestions of the editor and the reviewers as stated below. We highlighted the main changes to the text by red.
Reviewer 2
Point 1: The work is interesting, but it contains many scientific and editorial errors and cannot be published in its current form. The text should be corrected by a chemist. The main weaknesses are listed below:
- line 11, 73, the authors in the text write the Multiflora rose twice and omit Damascena.
- the authors use incorrect names of chemical compounds in the text, tables and figures. It is “exanal” and it should be “hexanal”, it is “trans-2-exenal” and it should be “trans -2-hexenal”, it is “trans-3-exen-1-ol” and it should be “trans-3-hexen-1-ol”, it is “trans-2- exen-1-ol” and it should be “trans-2-hexen-1-ol”, there is “alfa-pinene” and it should be “α-pinene”, there is “Citral_E” and it should be “(E)-citral”, there is “Citral_Z” and it should be “(Z)-citral”, is “3, methyl-1-butanol” should be “3-methyl-1-butanol” is “2-phenyethanol” and it should be “2-phenylethanol”
Point 1: we thank the reviewer. We modified the text, accordingly.
Point 2: Table 2. The authors give incorrect references to chemical compounds. For example, in publication 32 we will not find information about acetaldehyde, 3-methyl-1-butanol, 1-pentanol, (Z)-citral, (E)-citral, and in publication 34 we will not find information about trans-2-hexen-1ol. Identification of compounds on the basis of retention time only is insufficient, the values of the Kovats retention indices must be included in table 2.
Point 2: we checked the listed references, and we added more appropriate new references. We added a new column with Kovats retention indices.
Point 3: Chapter 2.2 does not contain significant information about the performed GC-MS analysis: chromatograph model, headspace automatic injector model, amount of sample injected, mass spectrometer parameters, MS library and so on. The given column temperature program is incorrect, it shows that the end is after 15 minutes and methyl eugenol, nerol, (E)-citral, (Z)-citral have longer retention times.
Point 3: we modified the Chapter 2.2 by adding more detailed information about the GC-MS analysis performed.
We remain available to clarify any issue or answer that Reviewers or Editors may raise.
Best regards,
Matteo Caser and
Valentina Scariot
Department of Agricultural, Forest and Food Sciences
University of Turin
Largo Paolo Braccini, 2
10095, Grugliasco (TO)
Italy
Phone number: +039-011/6708935
Fax number: +039-011/6708798
e-mail: matteo.caser@unito.it

Round 2
Reviewer 2 Report
The manuscript: horticulturae-1943260 “The contribution of volatile organic compounds (VOCs) emitted by petals and pollen to the scent of garden roses” cannot be published in its current form as it contains the errors listed below.
- Not all of the Kovats Retention Index values in Table 2 are correct, for example: 3-methyl-1-butanol, Trans-2-hexenal, Hexan-1-ol and so on.
- The authors did not disclose how the values of the retention indices were determined. RI rather than Ki is used for the retention index values.
- In the case of HS TriPlus 300, the full name of the manufacturer and address data were not given, as in the case of Agilent Technologies.
- In section 2.2, the data on time and temperature given are given with an accuracy inconsistent with generally accepted rules.
- Line 73 has not been corrected with the earlier guidelines.
- Table 4 is "A-pinene" should be "α-pinene".
Author Response
October 26th, 2022
To:
Editorial Office
Horticulturae
Dear Editorial Office,
We submit a revised version of the article “The contribution of volatile organic compounds (VOCs) emit-ted by petals and pollen to the scent of garden roses” by Matteo Caser and Valentina Scariot for publication in the journal “Horticulturae”.
We thank for the comments and suggestions that were very helpful to further improve clarity of the manuscript.
For the preparation of the revised manuscript, we followed all the comments and suggestions of the editor and the reviewer as stated below. We highlighted the main changes to the text by red.
Reviewer 2
Point 1: The manuscript: horticulturae-1943260 “The contribution of volatile organic compounds (VOCs) emitted by petals and pollen to the scent of garden roses” cannot be published in its current form as it contains the errors listed below.
Not all of the Kovats Retention Index values in Table 2 are correct, for example: 3-methyl-1-butanol, Trans-2-hexenal, Hexan-1-ol and so on.
Point 1: we thank the Reviewer for the suggestions. We checked the Kovats Retention Index for each compound on two online databases www.pubchem.com and www.flavornet.org and we added the corresponding values.
Point 2: The authors did not disclose how the values of the retention indices were determined. RI rather than Ki is used for the retention index values.
Point 2: we added the used references and we modified the text accordingly.
Point 3: In the case of HS TriPlus 300, the full name of the manufacturer and address data were not given, as in the case of Agilent Technologies.
Point 3: we added the requested information.
Point 4: In section 2.2, the data on time and temperature given are given with an accuracy inconsistent with generally accepted rules.
Point 4: we checked the data on time and temperature and we modified section 2.2, accordingly.
Point 5: Line 73 has not been corrected with the earlier guidelines.
Point 5: we modified the text, accordingly.
Point 6: Table 4 is "A-pinene" should be "α-pinene".
Point 6: we modified Table 4, accordingly.
We remain available to clarify any issue or answer that Reviewers or Editors may raise.
Best regards,
Matteo Caser and
Valentina Scariot
Department of Agricultural, Forest and Food Sciences
University of Turin
Largo Paolo Braccini, 2
10095, Grugliasco (TO)
Italy
Phone number: +039-011/6708935
Fax number: +039-011/6708798
e-mail: matteo.caser@unito.it
